# Mass-Sensitive Sensing of Melamine in Dairy Products with Molecularly Imprinted Polymers: Matrix Challenges

**DOI:** 10.3390/s19102366

**Published:** 2019-05-23

**Authors:** Martin Zeilinger, Hermann Sussitz, Wim Cuypers, Christoph Jungmann, Peter Lieberzeit

**Affiliations:** University of Vienna, Faculty of Chemistry, Department of Physical Chemistry, Währinger Straße 42, Vienna 1090, Austria; martin.zeilinger@gmx.net (M.Z.); Hermann.Franz.Sussitz@univie.ac.at (H.S.); Wim.Cuypers@univie.ac.at (W.C.); Christoph.Jungmann@univie.ac.at (C.J.)

**Keywords:** melamine, protein, bovine serum albumin (BSA), molecularly imprinted polymers (MIP), quartz crystal microbalances (QCM)

## Abstract

Food standards and quality control are important means to ensure public health. In the last decade, melamine has become a rather notorious example of food adulteration: Spiking products with low-cost melamine in order to feign high amino acid content exploits the lack in specificity of the established Kjeldahl method for determining organic nitrogen. This work discusses the responses of a sensor based on quartz crystal microbalances (QCM) coated with molecularly imprinted polymers (MIP) to detect melamine in real life matrices both in a selective and a sensitive manner. Experiments in pure milk revealed no significant sensor responses. However, sensor response increased to a frequency change of −30Hz after diluting the matrix ten times. Systematic evaluation of this effect by experiments in melamine solutions containing bovine serum albumin (BSA) and casein revealed that proteins noticeably influence sensor results. The signal of melamine in water (1600 mg/L) decreases to half of its initial value, if either 1% BSA or casein are present. Higher protein concentrations decrease sensor responses even further. This suggests significant interaction between the analyte and proteins in general. Follow-up experiments revealed that centrifugation of tagged serum samples results in a significant loss of sensor response, thereby further confirming the suspected interaction between protein and melamine.

## 1. Introduction

Melamine is a heterocyclic aromatic compound, which sees widespread use in the production of synthetic aminoplast resins and foams [1,2]. It has come under scrutiny after a series of scandals concerning the misuse of melamine in pet food as well as dairy products intended for consumption by infants [3,4]. The incentive of adding melamine to food products is the fact that it feigns higher than actual protein amount and thus product quality: Most standards to determine protein content of foodstuff make use of the Kjeldahl method [5], a quick, yet unspecific way of determining organic nitrogen, and using this value to assess protein contents. Melamine is nephrotoxic and drastically promotes the formation of kidney stones, especially in combination with cyanuric acid, its metabolite [6]. Together, they may lead to organ failure or, even worse, death [7]. The consequences of such food scandals are dramatic: In the US it is speculated that the contamination of pet food may have directly caused the death of several hundred animals with some sources even claiming figures in the low thousands [8]. Even worse, in China, roughly 300,000 infants were affected by the consumption of contaminated milk products [9]. Approximately 52,000 children had to be hospitalized and six eventually succumbed to organ failure [10]. According to the WHO, the incident was one of the largest in recent times [11]. It is, therefore, of fundamental interest to develop a sensor for fast and reliable detection of melamine in food products in order to guarantee and monitor their safety for consumption.

Current detection methods for melamine commonly rely on high performance liquid chromatography (HPLC) [12] and gas chromatography (GC) [13], often combined with mass spectrometry. However, there are only a few sensor systems available for melamine, which is a bit surprising. Some examples are: Garber et al. [14] assessed commercial enzyme-linked immunosorbent assays (ELISA) for detecting melamine in a highly sensitive manner. Lu et al. in 2017 compiled different methods for detecting melamine in food samples [15]. Biomimetic sensor systems proposed so far to detect melamine in milk and milk powders rely either on electrochemical approaches [16,17], or fluorescence/luminescence detection [18,19,20]. Furthermore, Li et al. [21] came up with a continuous method to use DNA loops for continuously monitoring melamine in flowing milk via electrochemical techniques. However, most of the electrochemical methods proposed use rather complex electrode systems; fluorescence techniques usually comprise of adding nanoparticle/nanocomposite solutions to samples, which makes them discontinuous.

To address such limitations, the work herein proposes a system comprising quartz crystal microbalances (QCMs) coated with molecularly imprinted polymers (MIP).

The QCM is a mass-sensitive transducer that exploits the piezoelectric effect found amongst others in quartz [22]. Originally discovered by the Curie brothers, the piezoelectric effect describes the formation of dipoles within piezoelectrically active materials upon exertion of mechanical stress [23,24]. Summarized over the entirety of the crystal, these dipoles make for a measurable voltage. The effect is reversible. Hence, the inverse effect may also take place: Applying alternating current results in the oscillation of a piezoelectric substrate. Given a certain geometry, substrate thickness, and crystal cut, a quartz substrate will resonate at a given frequency. In the case of mass deposition on the substrate surface, its effective thickness increases. This, in turn, results in a decrease of its resonance frequency. As frequencies can be measured with great accuracy, quartz microbalances with f_0_ = 10 MHz, as used as part of the research presented, are typically capable of sensing mass differences in the low pg region [25]. This makes them powerful tools in the field of rapid analysis.

Molecular imprinting is a concept aimed at creating fully synthetic, high-affinity receptors featuring spatially organized functional groups [26,27,28,29,30,31]. By polymerizing monomers in the presence of a template, cavities of a specific shape & form are generated which act as selective binding sites for the template. The working principle of molecular imprinting is thus similar to the key & lock principle found in enzyme–substrate interactions: Both lead to combined chemical and steric selectivity [32]. Molecularly imprinted materials see widespread use in preparative chemistry as well as in the role of robust, cost-efficient stand-ins for enzymes while gaining more and more importance in the development of cheap, dependable, and sensitive sensors [33,34].

## 2. Experimental Section

### 2.1. Materials and Samples

Sodium peroxodisulfate (NaPS), *N*,*N*′-(1,2-Dihydroxyethylene)bisacrylamide (DHEBA), and melamine were bought from Alfa Aesar, Methacrylic acid (MAA), BSA and casein from VWR and Merck, respectively. All chemicals were at least of analytical grade, deionized water was used throughout experiments. Milk (0.5 % fat) and natural whey (0.1 % fat) were purchased at a local supermarket.

### 2.2. Preparation of the MIP/NIP—QCM Sensors

For preparing MIP, 1.5 mg melamine–the template, 8 µL MAA, as well as 17 mg DHEBA were dissolved in 400 µL water by heating to 60 °C and sonicating the mixture. We chose MAA as a functional monomer because of its acidic functional group, which can interact with melamine through hydrogen bonding. In the next step, we added 5 mg NaPS, a water-soluble radical initiator, followed by keeping the mixture at a temperature of T = 60 °C in order to start polymerization. After approximately 15 min, the solution became slightly turbid, which indicated that it had reached the gel point. At that point, the oligomer is ready to be spin-coated onto a quartz crystal microbalance. NIP synthesis is the same as for the MIP, but without adding melamine. QCM substrates were generally coated at 2000 rpm using a commercial spin coater (“Spincoat G3P-8” by Specialty Coating Systems) and left to dry and harden at room temperature overnight. To remove the template, we flushed the sensors with distilled water for one hour. This is based on fluorescence experiments (excitation wavelength λ_ex_ = 260 nm, emission wavelength λ_em_ = 365 m, measured on PerkinElmer LS50B). After 30 min of washing, we could no longer detect melamine in the respective washing solutions.

### 2.3. Experimental Setup

The measurement setup used in the experiments consisted of the following elements: A custom-made oscillating circuit connected to a power supply, a frequency counter, and a measuring cell, as well as a personal computer with custom-made software for data readout and recording. We delivered all samples with a peristaltic pump (Ismatec MCP Process IP-65) at 1 mL/min. Each measurement comprised of first mounting the corresponding sensor in the measuring cell (see Figure 1 for the setup). Then, we equilibrated it in the respective matrix solutions (water, whey, or (skimmed) milk), either in stopped flow (milk and diluted milk), or at 1 mL/min (water and whey) until reaching stable baseline signal, as indicated by constant frequency readings. After reaching a stable frequency, we replaced the background solution by the respective sample. Depending on the sample matrix, experiments took place in stopped flow or at flow conditions as indicated before.

QCM sensors used herein rely on 10 MHz AT-cut quartz crystals (13.8 mm diameter, 168 µm thickness, purchased from Great Microtama Industries, Surabaya, Indonesia). We screen-printed the respective electrode structures (sample side: Two electrodes, 5 mm diameter, connected to each other and to the electrical ground, opposite side: Two electrodes, 4 mm, connected to the phase of the respective oscillator circuit) using brilliant gold paste (Heraeus GGP-2093, 12% gold). One electrode of each pair served as the measuring electrode comprising the MIP, while the other held the NIP. This reference electrode helped us compensate for unspecific effects of temperature, density, and ionic strength when measuring in the liquid phase.

## 3. Results and Discussion

Several melamine MIPs have already been described in the literature. Hence, we decided to use an MIP which had previously been developed for use in chromatographic separation but changed morphology from bulk particles to thin film [35]. Before applying a sensor to real-life matrices, one, of course, needs to characterize it to make sure that it indeed interacts with the analyte and yields useful signals. Figure 2 shows typical characterization measurements of melamine MIP-QCM sensor, namely both a sensor response pattern (A), and the corresponding sensor characteristic (B).

As can be seen in part Figure 2A, the sensors in question exhibit reversible, concentration-dependent effects for melamine concentrations between 1 mg/mL up to 3200 mg/mL melamine, the latter corresponding to a saturated aqueous solution. They also show favorable MIP/NIP ratios with the MIP layers generally displaying signals higher than the respective NIP: Corresponding imprinting factors are roughly five. The difference signal between MIP and NIP yields monotonous sensor characteristic in a melamine concentration range spanning from 1 to 3200 mg/L, i.e., until the end of the solubility range of melamine in water. The respective sensor signals—i.e., the differences between signals on MIP and NIP channel—went from −15 to −1250 Hz. Figure 2B displays this relation between the concentration of melamine and its related sensor effect. Considering both the signal drift of the baseline (roughly 10 Hz during the first five minutes before injecting the lowest sample concentration) and noise (±1 Hz), one can calculate the smallest discernible sensor effect of Δf = −14 Hz. This corresponds to the frequency shift at 1 mG/L melamine (Δf = −15 Hz) and leads to a limit of detection of LoD = 8 µM, which is a very appreciable value for a QCM sensor. Since MIP and NIP are chemically identical, the frequency difference can be attributed entirely to successful imprinting of melamine.

Keeping in mind the application, selectivity assessment focused on two compounds, namely glucose and BSA, the latter seeing widespread use as a model protein in investigations regarding general protein–polymer interaction [36]. Glucose shows no sensor signal at all, while BSA leads to frequency shifts that correspond to one-third of those for melamine. This clearly demonstrates that the sensor is useful to detect melamine in real matrices.

Figure 3 shows the QCM sensor responses obtained when exposing the sensor first to skimmed milk (0.5% fat) followed by the same milk spiked with 3200 mg/L melamine. Two aspects are immediately noticed: First, one can see extreme effects and noise during pumping the milk sample through the system. The exact reason for this is unknown, but has to do with the fact that milk is an emulsion. Anyway, this problem does no longer occur after stopping the flow. Second, in contrast to aqueous solutions, there is no sensor response for milk spiked with 3200 mg/L melamine. Only after diluting melamine-spiked milk with water by a factor of 10 was it possible to obtain a useful signal of around −20 Hz. This finding suggests that binding of the analyte to certain components of the sample matrix may significantly reduce the sensor signal.

To prove this claim further, we carried out measurements in whey. Compared to milk (0.5% fat, 3.5% protein, and 4.1% sugar), it contains significantly less fat and protein (0.1% fat and 0.6% protein). Figure 4 shows the corresponding sensor responses:

Evidently, the sensor responds favorably towards melamine with sensor effects of roughly −330 Hz and −35 Hz, respectively, for the two concentrations chosen (3200 mg/L and 320 mg/L melamine in whey). When comparing different media, as showcased in Table 1, the sensor response increases with decreasing amounts of fat and protein, indicating that melamine is either more soluble in fat or is, in fact, binding to milk proteins.

Comparing a 1:10 dilution of milk and undiluted whey seems to contradict this theory at first glance since milk samples result in lower melamine sensor response compared to whey. However, one needs to consider that diluted milk in total contains lower overall amounts of fat and protein. To prove our claim of melamine–protein interaction, we tested the sensor response in whey spiked with 1600 mg/L melamine and added different concentrations of casein, a family of phosphoproteins typically found in mammalian milk. As Figure 5 shows, the sensor response decreases significantly when adding casein to the sample matrix.

This demonstrates that the sensor response strongly depends on the protein concentration in solution: A higher amount of protein results in lower overall sensor response and vice versa. One can explain this by melamine binding to several proteins resulting in the formation of melamine–protein complexes. The sensor cannot detect these due to the immense size difference between free melamine and the complex. The latter is also highly soluble in fat, effectively making it inaccessible to the sensor. Literature commonly refers to centrifugation as a means of preparing a sample and overcoming the issues associated with the complicated milk matrix [26]. However, if melamine readily binds to protein, this would result in discarding a significant portion of the analyte with the protein precipitate, thereby decreasing sensor response, which may, in turn, result in incorrectly calculating the melamine concentration and limit of detection. To prove that melamine, in fact, binds to protein, we prepared two different whey samples: One aliquot was spiked with melamine (3200 mg/L) before being subjected to centrifugation. The other was spiked with the same concentration of melamine just after centrifugation. Then, we exposed MIP-coated QCM to both solutions. Figure 6 shows the outcome of one such QCM measurement.

The sensor response in the case of the serum spiked before centrifugation is significantly smaller than for the serum spiked after centrifugation: The frequency shifts differ by a factor of more than four. This leads to the conclusion that melamine binds to the protein and is removed from the sample solution in the course of centrifugation. Therefore, one has to be wary of using centrifugation as a method of sample preparation when measuring in real milk. Melamine, however, does not only bind to casein: QCM experiments (see Figure 7) also revealed substantial interaction with BSA, a standard protein used for testing.

Comparing the sensor response for melamine in the presence of BSA to the one measured in pure water, it immediately becomes apparent that the latter is much higher (−388 Hz vs. −1345 Hz). This clearly shows that melamine binds to different proteins. Table 2 summarizes the outcomes of protein–melamine interaction studies and shows that recorded sensor effects strongly depend on overall protein concentrations, for both BSA and casein. This is problematic for a range of sample preparation methods, such as filtration or size exclusion chromatography. In general, one, hence, has to keep in mind that melamine may be lost during sample preparation, thereby reducing the corresponding sensor responses and leading to possibly false negative results.

The lower LoD of the MIP-QCM sensor systems proposed herein is two to three orders of magnitude higher, than those of previously published electrochemical [16,17] and optical [18,19,20] systems (low µM range vs. nM/tens of nM). On the other hand, the upper limit of detection is at the saturation concentration of melamine, i.e., 2.5 mM solution in water, which is much higher than for previously reported sensors. This makes our system interesting for real-life applications: In the case of intentional adulteration of dairy, one would not expect very low concentrations, because those would not be useful to simulate higher protein content during Kjeldahl test. In that sense, a dynamic range covering the entire concentration window of melamine in water is more interesting than detecting extremely low concentrations. Second, the MIP-QCM sensor proposed herein is completely label-free and does not involve complex functionalizing of electrode surfaces, which makes it interesting for practical applications.

## 4. Conclusions

Overall, QCM-MIP sensors proved useful for detecting melamine in water in a concentration range spanning from as little as 1 mg/L up to a saturated solution (3200 mg/L), which in principle covers the detection range for revealing food adulteration. However, utilizing those sensors in pure milk, i.e., a realistic real-life matrix, led to no responses. Only results from diluted milk and whey lead to meaningful sensor signals. Systematic evaluation with two model proteins—BSA and casein—revealed that melamine readily binds to milk proteins. For food analysis and sensing, these results lead to two main conclusions: First, one has to be wary, especially when detecting low concentrations of melamine in food samples, because they may be the result of melamine absorbed to protein, which is, hence, not accessible to rapid analysis. Second, the sensor is nonetheless useful for direct screening of liquid samples: If indeed melamine had been added to feign higher protein content, it would lead to discernible sensor signals: The less protein is present, the higher the signal. Of course, the MIP-QCM sensor does not allow for quantifying melamine in such cases. It still is a useful tool for first screening of samples directly in situ: Natural milk and dairy products contain no melamine at all.

## Figures and Tables

**Figure 1 sensors-19-02366-f001:**
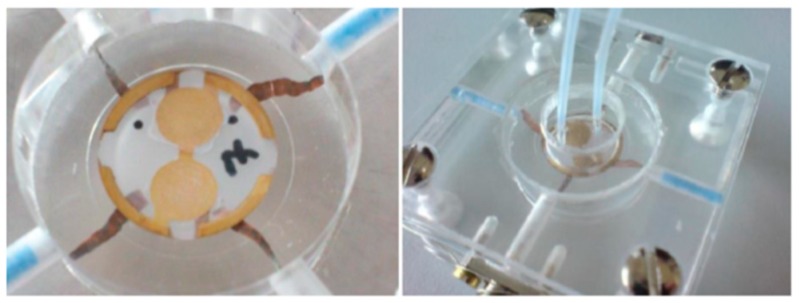
**Left**: Quartz crystal microbalances (QCM) positioned on the copper electrode wires. **Right**: Placement of the lid with inlet and outlet on top of the sensor.

**Figure 2 sensors-19-02366-f002:**
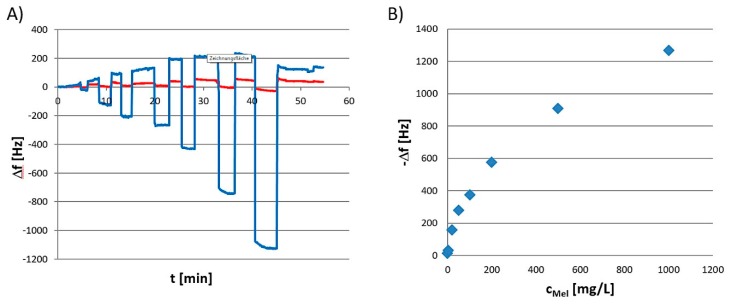
Sensor characterization of a melamine-imprinted sensor. (**A**) Sensor response pattern of dual-electrode QCM coated with MIP and NIP, respectively, when subjected to a number of different analyte concentrations ranging from 2–3200 mg/mL; (**B**) Corresponding sensor characteristic.

**Figure 3 sensors-19-02366-f003:**
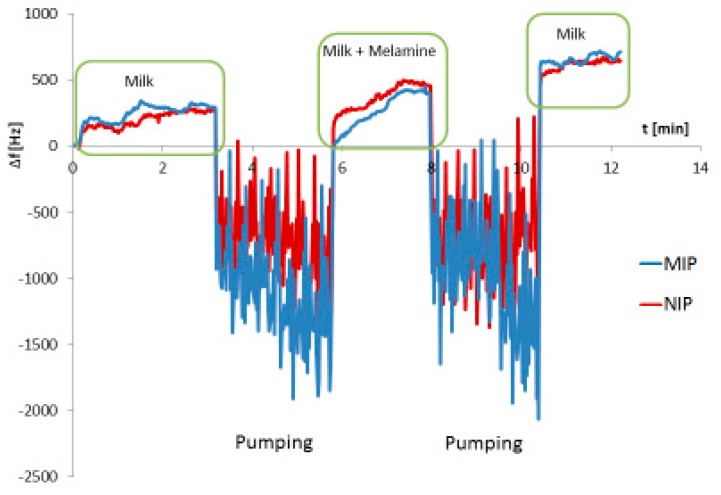
QCM measurement of melamine saturated milk (0.5% fat). Because of unknown matrix effects, there is no significant signal of melamine visible (green boxes). The enormous noise signals are due to the pumping of milk.

**Figure 4 sensors-19-02366-f004:**
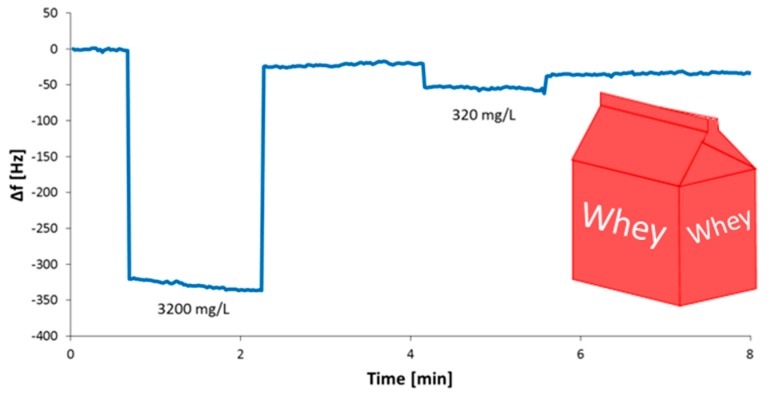
Sensor response for melamine measured in whey: ~−330 Hz at 3200 mg/L, ~−35 Hz at 320 mg/L.

**Figure 5 sensors-19-02366-f005:**
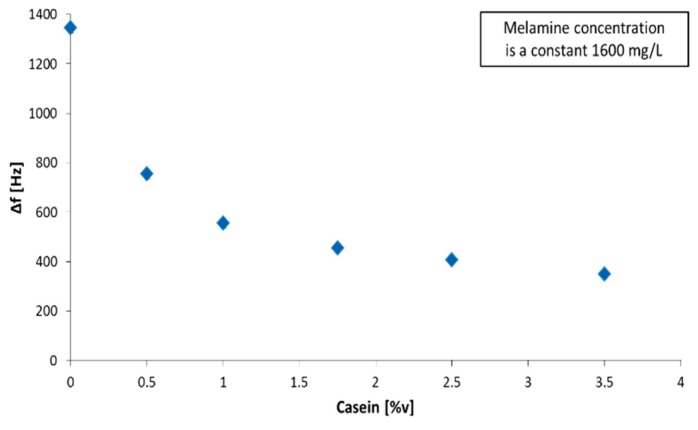
Sensor responses at a constant concentration of 1600 mg/L melamine, but at varying casein content.

**Figure 6 sensors-19-02366-f006:**
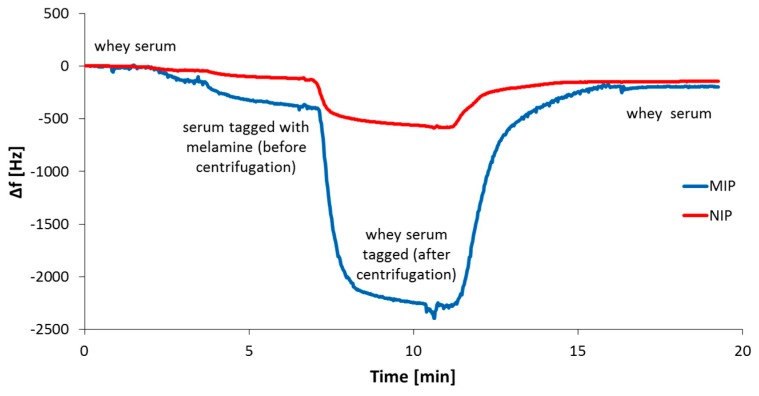
Centrifugation as a means of sample preparation: Sensor effects of serum aliquots tagged with melamine before and after centrifugation.

**Figure 7 sensors-19-02366-f007:**
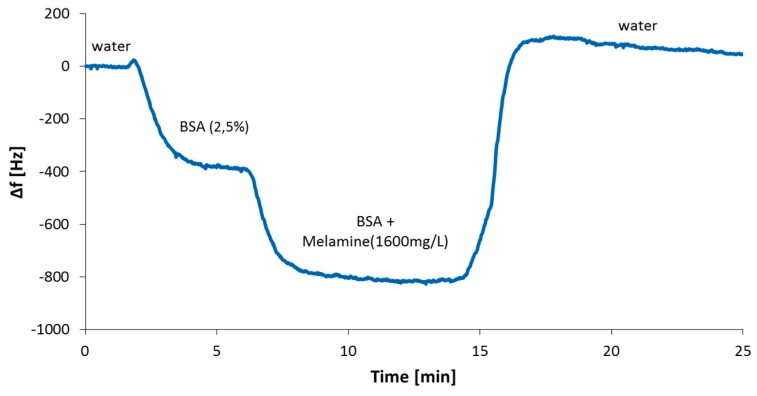
QCM measurement showing the reduced sensor effect upon injection of melamine (1600 mg/L) when measuring in a matrix spiked with BSA.

**Table 1 sensors-19-02366-t001:** Sensor responses in different sample matrices at a constant melamine concentration of 3200 mg/L.

Media	% Fat	% Protein	% Sugar	Signal *[Hz]
Water	0	0	0	1150
Whey 1:10	0.01	0.06	0.41	950
Milk 1:10	0.05	0.35	0.49	30
Whey **	0.1	0.6	4.1	370
Milk **	0.5	3.5	4.9	0

* 3200 mg melamine, ** values given in (*w*/*w* %).

**Table 2 sensors-19-02366-t002:** Sensor response dependency on protein concentration at a constant melamine concentration level of 1600 mg/L.

Protein Concentration [%]	BSA/Melamine [−Hz]	Casein/Melamine [−Hz]
0.0	970	1345
0.5	880	754
1	640	555
1.75	430	454
2.5	388	407
3.5	239	348

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
