# Peer review of "Mass-Sensitive Sensing of Melamine in Dairy Products with Molecularly Imprinted Polymers: Matrix Challenges"

_sensors, 2019, doi:10.3390/s19102366_

Round 1
Reviewer 1 Report
The manuscript sensors-474866 entitled "Mass-sensitive sensing of melamine in dairy products with molecularly imprinted polymers: Matrix challenges" authored by M. Zeilinger , H. Sussitz , W. Cuypers , Ch. Jungmann , P.A. Lieberzeit puts emphasis on important problem of bindig analyte to proteins present in the sample and thus causing false negative detemination/detection results. Up to now this issue was not much discussed by authors devising chemosesors. Therefore, this manuscript may be interesting for broad audience of researchers working in this particular field.
Author Response
We thank the reviewer for his positive opinion on our manuscript. During review, we have also edited for language, where necessary, and hope that we could improve the manuscript. The reviewer did not suggest any further changes.
Reviewer 2 Report
The manuscript aimed to demonstrate the sensitive detection of melamine using molecularly-imprinted polymer sensors and tried to translate the technology to spiked real milk samples. In the first part, there were a lot of missing vital information when the authors were fabricating a melamine MIP sensor. Here are some of these missing points:
· Did melamine remain stable after heating the solution at 60 Deg C? There should have been a parallel experiment investigating whether the structure of the analyte was maintained so that we can assume that the actual conformation of melamine was imprinted in the polymer.
· How was the melamine extracted from the initial analyte-polymer matrix? In addition, the incorporation and removal of the analyte should have been well characterized using various spectroscopic techniques.
· In the experimental section, it is unclear which environment was used in the QCM measurements. Were the measurements initially stabilized in dry, ambient conditions and the analyte was just injected? Or, were they initially stabilized in aqueous conditions in flow? What’s the flow rate?
· Was the QCM setup assembled in house or was this purchased? If it’s not purchased as a ready-to-use instrument, it would help if a schematic of the setup was shown.
· In Figure 1, why was the QCM baseline signal increasing before and after exposure to melamine? Did some of the polymer matrix get removed after the melamine exposure?
The second part wherein the sensor was tested against real milk samples were much more problematic. In the manuscript, the authors fleshed out all the things that may be interesting but ultimately did not work. These information is typically included in the supplementary material. It is expected that by including this section, that the authors have figured out a way to make the sensor work in milk samples. If not, then the whole section is not sufficiently interesting to be published. Overall, the only thing that somehow worked was the first part with the detection of melamine in an idealized environment. However, this has been demonstrated multiple times using different versions of MIP, which makes that portion not novel anymore. Hence, with all of these reasons, I do not recommend accepting this manuscript for submission.
Author Response
We are grateful to the reviewer for his in-deep, constructive assessment of the paper that substantially helped us improving the manuscript. Please find our responses in an itemized way, with the original reviewer comments printed in italics and our responses in plain text. We have marked all changes to the manuscript in yellow color.
· Did melamine remain stable after heating the solution at 60 Deg C? There should have been a parallel experiment investigating whether the structure of the analyte was maintained so that we can assume that the actual conformation of melamine was imprinted in the polymer.
We did not test this, but there is no reason to believe that the conformation of melamine changes at 60°C: it is a small, heteroaromatic molecule (M=126 g/mol) bearing three primary amino groups. Therefore, rotation of those NH2 groups around their binding axes is the only freedom of movement in the molecule, because the aromatic system is very rigid. Furthermore, 60°C is very far from the melting point of melamine (345°C).
· How was the melamine extracted from the initial analyte-polymer matrix? In addition, the incorporation and removal of the analyte should have been well characterized using various spectroscopic techniques.
It was extraced by washing with water controlled by fluorescence assessment. We agree to the reviewer that this information had been missing and included it to the manuscript. However, we disagree, that "various" methods would be necessary for characterizing this process.
· In the experimental section, it is unclear which environment was used in the QCM measurements. Were the measurements initially stabilized in dry, ambient conditions and the analyte was just injected? Or, were they initially stabilized in aqueous conditions in flow? What’s the flow rate?
This information indeed was missing from the manuscript. We added it to the experimental section.
· Was the QCM setup assembled in house or was this purchased? If it’s not purchased as a ready-to-use instrument, it would help if a schematic of the setup was shown.
In fact, it is assembled in-house. We included this information into the manuscript and added a figure of the cell setup.
· In Figure 1, why was the QCM baseline signal increasing before and after exposure to melamine? Did some of the polymer matrix get removed after the melamine exposure?
This is most probably due to slight loss of material for freshly prepared sensors. We have included the respective information into the manuscript (discussion of Fig. 2A).
The second part wherein the sensor was tested against real milk samples were much more problematic. In the manuscript, the authors fleshed out all the things that may be interesting but ultimately did not work. These information is typically included in the supplementary material. It is expected that by including this section, that the authors have figured out a way to make the sensor work in milk samples. If not, then the whole section is not sufficiently interesting to be published.
In this point we strongly disagree with the reviewer: the whole point of the paper is to show that the current analytical approach for detecing melamine in milk samples needs to be viewed in a critical manner, because proteins - which are inevitably present in milk - adsorb the analyte and lead to substantially decreased - or even false negative - effects. However, our in-situ measurements with different samples systematically show, where the melamine goes to. In that sense we are convinced that the present manuscript is substantially novel. So it is not about "fleshing out things .... that did not work", but about showing, where the analytical problem is. However, we obviously did not get this point through in the manuscript, so we explicitly included this information, for instance to conclusions. Looking at the three other review reports, this also seems to be their understanding.
Overall, the only thing that somehow worked was the first part with the detection of melamine in an idealized environment. However, this has been demonstrated multiple times using different versions of MIP, which makes that portion not novel anymore.
We fully agree to the reviewer, that showing the results of a melamine MIP in an indealized environment is not quite novel. However, as mentioned above, the whole point of our manuscript is "searching for the reason", why the sensor works perfectly at idelaized conditions, but not in real-life matrices. To the best of our knowledge, there are hardly any examples in literature of using MIP-based sensors in real-life systems such as milk and whey. The manuscript probably was not clear enough on that, so we amended it accordingly.
Reviewer 3 Report
This is a very good paper on important topic. It is important for us to have high quality food products and, thus, sensing if its quality is not adequate is very important. The work was performed in a highly professional manner. I have just one remark. Section 3 should be renamed as Results and Discussion and section 4 to Conclusions. This section should be shortened and make the composition harder. Otherwise, I recommend this work for publication.
Author Response
We very much appreciate the reviewer's comments and thank for his very postive assessment.
As suggested, we renamed section 3 to "Results and Discussion" and section 4 to "Conclusions". Furthermore, we edited section 4 to make it more concise.
We also proof-read the paper once more to correct for language.
All changes to the text are marked in yellow.
Reviewer 4 Report
The paper describes a MIP based QCM sensor for the detection of melamine.
Before I can recommend the publication, the following points should be addressed.
Introduction does not reflect the state of the art for melamine based sensors. The authors should further discuss the previous methods for the detection of melamine in terms of MIPs and biosensors.
Why was Melamine, sodium peroxodisulfate used as the template rather than just melamine?
Figure 1 What was the difference between the two QCM sensorgrams. The authors should explain this for reader clarity.
The limit of detection should be calculated for the sensor.
The authors failed to provide a conclusion in their manuscript.
How does the described sensor compare to other methods?
Line 183 Figure not in bold
Author Response
We thank the reviewer for his constructive comments that have helped us substantially improving the manuscript. Please find our responses in an itemized way, with the original reviewer comments printed in italics and our responses in plain text. We have marked all changes to the manuscript in yellow color.
Introduction does not reflect the state of the art for melamine based sensors. The authors should further discuss the previous methods for the detection of melamine in terms of MIPs and biosensors.
We have included a range of references on melamine sensors including a review article from 2017 (new references no. 12-20)
Why was Melamine, sodium peroxodisulfate used as the template rather than just melamine?
Actually, this is a misunderstanding: sodium peroxodisulfate is not the template, but the initiator. We agree that the text was not clear in this point and corrected accordingly.
Figure 1 What was the difference between the two QCM sensorgrams. The authors should explain this for reader clarity.
In fact, they are measurements originating from two different QCM. In one case, we flushed with water between analyte exposures, in the other case not. We included this information to the manuscript and discuss the differences. However, as this obviously leads to confusion, we show only one sensorgram in the revised version and add the sensor characteristic. Quantitative differences between the two curves result from the fact that layer heights are different.
The limit of detection should be calculated for the sensor.
We agree with the referee and added this information together with the new version of Fig. 2.
The authors failed to provide a conclusion in their manuscript.
Indeed. A second also reviewer pointed out this mistake, which we of course corrected.
How does the described sensor compare to other methods?
We included this aspect into the our Discussion section. However, the main point of this work has not been to "outshine" other sensors in terms of LoD or analytical behavior, but to demonstrate the difficulties when carrying out measurements in real-life samples. Although several other papers demonstrate the same, those used comparably low melamine concentrations, which are not interesting in terms of adulterating dairy.
Line 183 Figure not in bold
We have corrected this.
Round 2
Reviewer 2 Report
I appreciate that the authors carefully considered my initial comments and have updated the manuscript accordingly. The changes incorporated in the manuscript have greatly improved the clarity of the story and, at the same time, sorted out a lot of the missing technical information that researchers in MIP easily notice and question upon seeing new MIP-related papers. With these, I am satisfied with the current version and recommend publishing in this journal.